# Emotional Intelligence and Its Relationship to Basic Psychological Needs: A Structural Equation Model for Student Athletes

**DOI:** 10.3390/ijerph191710687

**Published:** 2022-08-27

**Authors:** Isabel Mercader-Rubio, Nieves Gutiérrez Ángel, Nieves Fátima Oropesa Ruiz, José Juan Carrión Martínez

**Affiliations:** Departamento de Psicología, Universidad de Almería, 04120 Almería, Spain

**Keywords:** emotional intelligence, athletes, basic psychological needs

## Abstract

This paper analyses the relationship between emotional intelligence (attention, clarity and repair) and motivational mediators (relationships, autonomy and perceived competence) of students of different degrees related to physical activity and sports science. A structural equation model was estimated with a sample of university students. The results of the model are representative of this case study only and are not very generalizable due to the sample size. In any case, the results obtained show that emotional intelligence predicts the satisfaction of psychological needs for autonomy and competence in student athletes. Therefore, these demonstrations point to a relationship between both variables and highlight the importance of addressing this type of content in university classrooms in order to increase the positive effects on the psychosocial development and personal well-being of students.

## 1. Introduction

Emotional intelligence is one of the most innovative psychological constructs within sports psychology in the 21st century and, currently, the lines of research on it correspond to a booming topic [1,2,3,4]. There are two main theories that seek to understand the concept of emotional intelligence: the trait model [5] and the ability model [6]. Briefly, they differ in two fundamental aspects: in the dimensions of emotional intelligence that make up each of them, and in the measurement of the same [6]; that is, in the concept itself that each of them provides, in the objectives they pursue, the assessment methods and the dimensions that make up emotional intelligence [7]. However, there are also a series of common criteria in all models of emotional intelligence, which allude to the triple purpose of the same, which is none other than the identification, mastery and use, in an optimal way adapted to the context, of one’s own emotions. Therefore, the most significant differences between some models and others lie in the number of competencies and skills established and in the formulation of the concept [8]. In this paper we opt for the second model composed of different sections or branches, where each of them has a series of skills: (1) emotional perception is the ability to identify and recognize one’s own and others’ feelings, and it implies, therefore, an interest in and knowledge of different signs about the expression, sensations and sincerity of emotions; (2) emotional understanding is about knowing, classifying and examining emotions, retrospectively, both one’s own and others’; (3) emotional regulation is about capturing, analyzing and reflecting on emotions in order to make the most of them, both interpersonal and intrapersonal emotions. In this sense, such instruments correspond to the first formal attempt to measure emotional intelligence, and they are still widely used today due to their advantages, among which stand out their easy and quick collection and analysis of information, as well as the consideration of introspective processes [9]. However, we should not forget their notorious disadvantages (biases of various kinds or measurement overlaps with other measures), as a result of which new instruments for the assessment of emotional intelligence have emerged [10].

However, it should be noted that despite the existence of these two major theories, and despite their discrepancies, both share several common features, such as the fact that emotions are considered predictors of positive adaptive behaviors [11], such as self-determination and motivation. Self-determination theory [12] indicates the motives that lead a person to initiate and maintain a behavior [13]. Within this theory we also find the importance of three basic psychological needs: competence, autonomy and relatedness to others [14]. Thus, competence refers to the way in which a person relates to and copes with everyday life effectively and confidently [12], while autonomy corresponds to the decisions that the person makes, and relatedness refers to the interpersonal relationships that the person establishes [8,9,10,11,12,13,14,15,16,17]. In this sense, we can affirm that such a theory alludes, therefore, to motivation and its types [15], understanding motivation as the engine that drives a person to undertake an action [16], and differentiating between intrinsic motivation (which corresponds to the enjoyment of the task itself, causing pleasure and psychological well-being) and extrinsic motivation (in which motivation comes from an external factor) [17]. Combining both ideas, we can establish the existence of a correspondence between the two (satisfaction of basic psychological needs and motivation) so that the greater the satisfaction of these needs, the more a person’s actions are considered to be produced by a person, i.e., by intrinsic motivation [18], and, conversely, the lower this satisfaction, the more a person’s actions are produced by extrinsic motivation [19]. Therefore, in this work we have used the scale of motivational mediators in sport (EMMD) to measure the satisfaction of basic psychological needs in the physical–sports environment, based on the theory of self-determination. The most common type of motivation in sport is extrinsic motivation, which in this field refers to external incentives, the avoidance of guilt, the consideration of the activity as important but not pleasant or because sport itself corresponds to a healthy and active lifestyle, also called integrated regulation [20,21,22]. Therefore, this work is framed within others that have also aimed to investigate the theory of self-determination in athletes [23,24,25,26,27,28,29].

Thus, through this research work we intend to analyze whether emotional intelligence (attention to feelings, emotional clarity and emotion repair) predicts the satisfaction of those from the theory of self-determination, understood as motivational mediators in the field of sports. Therefore, the research questions we intend to answer are: Is self-determination theory a motivational mediator? Is emotional intelligence a predictor of basic psychological needs? What kind of relationship exists between emotional intelligence and self-determination theory, understood as motivational mediators in the field of sports? In this sense, we must also add the fact that a new sample has been chosen, in this case, athletes in training from different degrees related to the sciences of physical activity and sport, due to the researchers’ concern to know whether these issues are taken into account and addressed in their training from the beginning of the athletes’ professional training.

The hypotheses of this work are: 

**Hypothesis** **1.***Emotional intelligence is a predictor variable of positive adaptive behaviors*.

**Hypothesis** **2.***Emotional intelligence is a predictor variable of the way in which a person relates to and copes with daily life effectively and confidently, with a direct and positive relationship*.

**Hypothesis** **3.***Emotional intelligence is a predictor variable of the decisions a person makes, with a direct and positive relationship*.

**Hypothesis** **4.***Emotional intelligence is a predictor variable of the interpersonal relationships that the person establishes, with a direct and positive relationship*.

## 2. Materials and Methods

The method used was correlational, corresponding to an ex post facto design, and of a retrospective and comparative nature since the dimensions of emotional intelligence are compared with other types of variables, in this case dependent variables, corresponding to competence, autonomy and relationships with others. In addition, a structural equation model on the satisfaction of basic psychological needs (competence, autonomy and relationship) was carried out, with emotional intelligence acting as a predictor.

### Participants

The sample consisted of 165 undergraduate and master’s degree students related to physical activity and sports science of the Spanish Public University. The average of the age of the sample was 20.33 years, with a standard deviation SD = 3.44. Regarding gender, 70.9% (*n* = 117) were male and 27.9% (*n* = 46) were female. The sample size was calculated using an online calculator (Soper, 2022), which allows for an a priori power analysis, taking into account the proposed SEM model. Thus, based on six observable variables, one latent variable, with an anticipated effect size of 0.30, a desired probability of 0.05 and a power level of 0.95, the minimum recommended sample size was 200 cases, so the number of participants in our study was close to the suggested number of people. After informing the students and obtaining their consent to participate in the study, the questionnaire was administered to the four courses of the degree in physical activity and sports science and to the students of the master’s degree in teaching (specialization in physical education) and the master’s degree in sports science research.

## 3. Results

### 3.1. Instruments

The instruments used in this work were the following: The TMMS-24 [30] corresponds to a kind of sub model on emotional intelligence created from the model of Salovey and Mayer [6], which some authors call the Spanish version of the model of [6]. For these authors, emotional intelligence is composed of three dimensions, such as perception, understanding and regulation of emotions, through a Likert-type scale [31]. For this work, Cronbach’s alpha = 0.84 was obtained. Moreover, this shows the high reliability for each dimension (perception, α = 0.90; clarity, α = 0.90; regulation at α = 0.86). Other studies have shown that the questionnaire has adequate test–retest reliability: perception = 0.60; comprehension = 0.70 and regulation = 0.83 [32]. In this paper, the TMMS-24 [9] was used as a measurement instrument in line with the theoretical model of emotional intelligence [10]. This instrument is a self-report measure of perceived emotional intelligence, i.e., a person’s self-awareness of his or her own emotional capabilities: attention to feelings, emotional clarity and emotion repair. The sport motivational mediators scale [24] measures the satisfaction of basic psychological needs in the sport context [24] through a Likert-type scale. It is composed of 23 items grouped into 3 factors: (a) autonomy: eight items; (b) perceived competence: seven items; (c) relationship with others: eight items. A Cronbach’s alpha = 0.71 was obtained for this work. The reliability of the original instrument is Cronbach’s alpha = 0.75 for the first factor, Cronbach’s alpha = 0.76 for the second, and Cronbach’s alpha = 0.71 for the third [24].

### 3.2. Data Analysis

The data analyses used in this study were descriptive statistics (mean, standard deviation and bivariate correlations), reliability analysis and structural equation modelling (SEM) to test the relationships established in the hypothesized model [33,34]. To accept or reject the proposed model, a set of fit indices were taken into account [35]: TLI (Tucker–Lewis index), SRMR (Standardized Mean Root Square Residual) and RMSEA (Root Mean Square Error of Approximation). Thus, the fit indices are: TLI value above 0.95, SRMR values below 0.06, and RMSEA with values below 0.08 (as can be seen in Table 1). These analyses were carried out using SPSS version 26 (IBM, Armonk, NY, USA) and R statistical analysis software (R Foundation for Statistical Computing, Vienna, Austria).

The relationships between each of the dimensions of EQ: emotional attention, emotional clarity and emotional regulation (EQ/CE/ERM) and their relationships with competence, autonomy and relatedness were analyzed. As can be seen in Table 2, the correlations between the study variables were positive in most cases, reflecting the reciprocity between the study variables. 

Table 2 shows the means, standard deviations and Pearson correlations for the variables studied. Pearson’s correlation analyses revealed, on the one hand, that competence correlated significantly with the dimension of emotional regulation (r = –0.222, *p* < 0.01), with a large effect size, as did autonomy with the dimensions of emotional regulation (r = –0.304, *p* < 0.01) and emotional clarity (r = –0.233, *p* < 0.01), while the relationship correlated significantly with the emotional clarity dimension (r = –0.236, *p* < 0.01) (see Figure 1).

The relationships established in the structural equation model are specified below: (a)Emotional intelligence and autonomy, understood as the own decisions made by a person, in our case an athlete, were positively correlated (=0.89, *p* < 0.001). This allows us to explain that, in this case, emotional intelligence is a predictor of decision making and, therefore, the presence of this variable explains the existence of the other variable.(b)Emotional intelligence and competence, understood as the way in which an athlete relates to and deals with his or her day-to-day life with efficiency and confidence, were positively related (=0.40, *p* < 0.001). These results show that emotional intelligence also predicts how a person copes with the circumstances that arise in the context in which he or she operates.(c)Emotional intelligence and relatedness were not positively related. The results found do not establish that emotional intelligence is a predictor variable of the relationship and, therefore, the presence of emotional intelligence does not correspond to the occurrence of this variable.

From these results we can establish the following: 

**Hypothesis** **1.***Emotional intelligence is a predictor of positive adaptive behaviours. This hypothesis is partially fulfilled*.

**Hypothesis** **2.***Emotional intelligence is a predictor of how a person relates to and copes with everyday life effectively and confidently, with a direct and positive relationship. This hypothesis is not fulfilled*.

**Hypothesis** **3.***Emotional intelligence is a predictor of the decisions a person makes, with a direct and positive relationship. This hypothesis is fully satisfied*.

**Hypothesis** **4.***Emotional intelligence is a predictor variable of the interpersonal relationships that the person establishes, with a direct and positive relationship. This hypothesis is fully satisfied*.

## 4. Discussion

The aim of the present study was to analyze whether emotional intelligence (attention to feelings, emotional clarity and emotion repair) predicts motivation in athletes, based on self-determination theory. It should be noted that the topic of self-determination theory is present in a large number of research studies related to the field of sport [24,36,37,38]. The aim was to answer the fact as to whether self-determination theory is a motivational mediator and whether emotional intelligence is a predictor of basic psychological needs, and, what kind of relationship exists between emotional intelligence and self-determination theory in the field of sport. As with other work previously developed, this research mainly focused on the study of the relationships between the satisfaction of basic psychological needs and other types of psychological variables, such as coping with sport tasks [39], physical self-concept [24,40,41,42] or anxiety before decision making [43]. However, none of these studies focused on the importance of the emotional training of athletes, as little attention has been paid to how emotional intelligence affects athletes in the satisfaction of basic psychological needs such as autonomy, competence and relatedness [43,44]. 

However, the results obtained in this work demonstrate, in our case, that emotional intelligence favors the positive development of aspects such as competence and decision making in athletes. This is along the same lines as other studies that have attempted to determine the relationship between self-determination theory and emotional intelligence, the results of which also corroborate that high levels of emotional intelligence are associated with greater satisfaction of the three basic psychological needs [45,46,47,48,49]. 

This leads us to think about the necessary emotional training that athletes should have from their initial training. We should also point out that the results obtained show the fact that emotional intelligence did not predict the satisfaction of relational needs in athletes. Therefore, we consider that it would be convenient to replicate the study with larger samples in order to contrast these findings, as this is one of the current limitations of the work. 

## 5. Conclusions

Finally, we would like to highlight the importance of the need to address psychological aspects such as emotional intelligence, which is not only cognitive in nature, but also psychological and emotional, both in terms of the competencies of the athlete in the context in which he or she plays, as well as in decision making, interpersonal relationships, mainly in team sports, and group cohesion and the feeling of belonging to the group, all of which are fundamental for favorable social development. 

Future lines of research will aim to determine whether there are differences according to the type of sport practiced, as well as to expand the sample size, as this has been one of the limitations of this study. Thus, future studies will attempt to analyze the differences according to the degree of professionalization of each sport practiced by the future participants.

## Figures and Tables

**Figure 1 ijerph-19-10687-f001:**
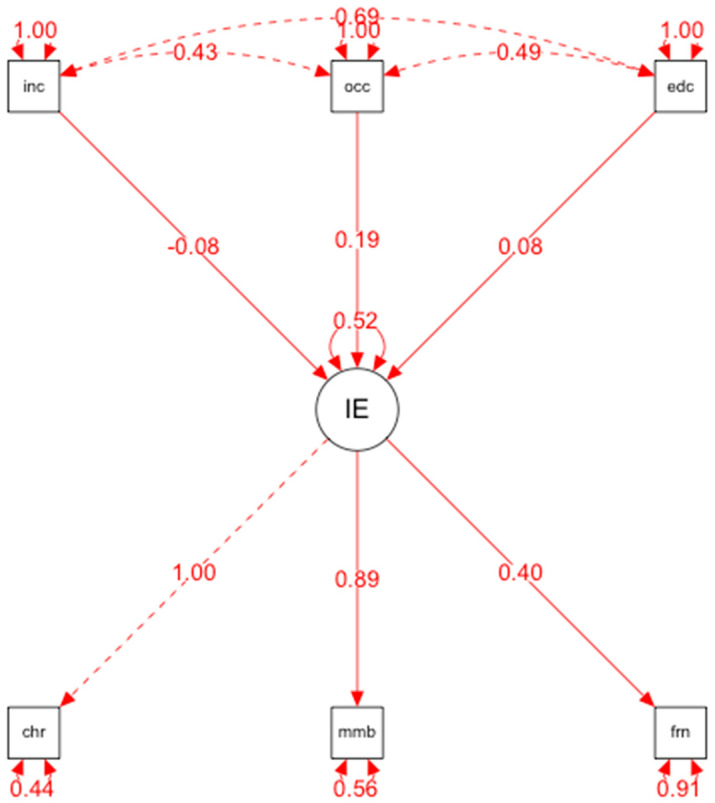
Structural equational modeling. INC: emotional attention. OCC: emotional clarity. EDC: emotional regulation. CHR: relation. MMB: autonomy. FRM: competence.

**Table 1 ijerph-19-10687-t001:** Overall model fit indices.

Index	Value	Evaluation of Fit
χ^2^/gl	0.001	Good
Root Mean Square Error of Approximation (RMSEA)	0.070	Good
Root Mean Square Residual (RMR)	0.032	Good
Comparative Fit Index (CFI)	0.944	Good

**Table 2 ijerph-19-10687-t002:** Preliminary Analysis.

	1	2	3	4	5	6
1. Emotional attention (AE)						
2. Emotional clarity (CE)		0.127	0.263 **	0.078	0.233 **	0.068
3. Emotional regulation (ERM)			0.508 **	0.246 **	0.304 **	0.222 **
4. Relation				0.216 **	0.315 **	0.216 **
5. Autonomy					0.532 **	0.478 **
6. Competence						0.296 **

Note. ** *p* < 0.01.

## Data Availability

Not applicable.

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
