# Peer review of "Emotional Intelligence and Its Relationship to Basic Psychological Needs: A Structural Equation Model for Student Athletes"

_ijerph, 2022, doi:10.3390/ijerph191710687_

Round 1

Reviewer 1 Report (Previous Reviewer 1)

I commend the authors on a sound and thorough revision. I'd suggest revising the hypotheses. H1 should read "is" rather than "as" to have parity with other hypotheses. But this change may be irrelevant, because I think all hypotheses would be better stated as "[Independent variable name] will exhibit a significant association with [dependent variable name]." I'd personally avoid "predict" or "predictor" as causal order can be a challenge, though I know that's commonly done (and I've used predictive language from time to time). If the authors prefer to retain predictive language given SEM, I'm good with what they decide but H1 then needs minor revision. Well done. 

Author Response

Thank you very much for your contributions.

We prefer to keep the predictive language given SEM, and H1 has been reformulated

Reviewer 2 Report (Previous Reviewer 2)

The authors addressed all the changes that were required.

Author Response

Thank you very much for your contributions.

This manuscript is a resubmission of an earlier submission. The following is a list of the peer review reports and author responses from that submission.

Round 1

Reviewer 1 Report

This paper on emotional intelligence among student athletes has promise and offers interesting preliminary results. However, it is quite deficient in its present form. Careful and detailed attention should be given to the following considerations.

1. The authors mention the trait model and ability model for emotional intelligence but no attention is given to the former. If both are introduced, the distinctions between them need to be addressed and then the choice to follow or test one model requires a compelling justification.

2. What are the limits of an instrument predicated on self-reports of emotional intelligence? Are we not better off with more objective measures? A defense of the instrument is needed.

3. The introduction of the paper digs into the literature review with no additional prior research presented. I believe the literature review should be its own section. The introduction should be revised to provide general background, clearly state the research problem, and then justify the significance of the study. These items are missing from the current introduction. Concerning significance, more needs to be said about student athletes. Is this a new or unique population? Please make a clear and compelling case for your specific investigation. That is not done.

4. The paper needs careful proofreading. Doesn't this statement come from the submission guidelines and template? Why is it in the paper? "This section may be divided by subheadings. It should provide a concise and precise description of the experimental results, their interpretation, as well as the experimental conclusions that can be drawn." I hope this doesn't sound harsh, but this type of oversight can raise questions about the care the authors have invested in any facet of this study. Also, I typically see a space on either side of an equals sign (x = 1.23). This is inconsistently done in the paper. Careful revisions are needed to inspire confidence in the reviewers and readers.  

5. Cociente is quotient, right? Translate into English, please.

6. Why are there commas before the Table 2 statistics?

7. The results would be more easily understood if preceded by hypotheses placed after the lit review.

8. The Discussion and Conclusions section is unduly terse. If a prospective reader wanted to see a restatement of the research question and summary of the key findings along with implications, limitations, and directions for future research, this is where they would look. Most, if not all, of this is missing. 

A lot of work is needed for manuscript to be considered further if it is granted a revise and resubmit. 

Author Response

  1. The authors mention the trait model and the skills model for emotional intelligence, but do not pay attention to the former. If both are introduced, it is necessary to address the distinctions between them and then the choice to follow or test a model requires convincing justification.

10 lines (line 73-82) have been included where the differences and similarities between both ways of understanding emotional intelligence and the conceptualizations of each model are addressed.

  1. What are the limits of an instrument based on self-reports of emotional intelligence? Aren't we better off with more objective measures? A defense of the instrument is needed.

10 lines (line 90-211) have been included where the reasons for choice, advantages and disadvantages of this type of instrument are identified.

  1. The introduction of the article delves into the literature review without presenting additional previous research. I think the literature review should be its own section.

A total of five new references have been added in the introduction as an allusion to previous research related to this topic.

  1. The introduction should be reviewed to provide general background, clearly state the research problem, and then justify the importance of the study. These elements are missing from the current introduction.

The introduction has been revised, restructured and reworked. Incorporating a total of three research questions. As well as the influence of emotional intelligence in the sports field, on sports performance, and the self-concept of athletes.

  1. As for the importance, it is necessary to say more about student-athletes. Is this a new or unique population? Present a clear and compelling case for your specific research. That is not done.

Sample selection criteria are argued (lines 203-207)

  1. The document needs careful review. Doesn't this statement come from the guidelines and presentation template? Why is it in the newspaper? "This section can be divided into subtitles.

The document has been carefully revised in its entirety

  1. It should provide a concise and accurate description of the experimental results, their interpretation, as well as the experimental conclusions that can be drawn."

The results have been written providing a description of them.

  1. Also, I usually see a space on each side of an equal sign (x = 1.23). This is done inconsistently in the document. Careful reviews are needed to inspire confidence in reviewers and readers.

The document has been carefully revised in its entirety

  1. Quotient is quotient, isn't it? Translate into English please.

It has been deleted and only the formula has been left.

  1. Why are there commas before the statistics in Table 2?

Commas have been replaced by periods.

  1. The results would be more easily understandable if they were preceded by hypotheses placed after the literature review.

A total of four hypotheses have been added in the introduction (line 210-219).

This has also been added in the interpretation of the results (lines 351-370)

  1. The Discussion and Conclusions section is unduly concise. If a prospective reader wanted to see a reformulation of the research question and a summary of key findings along with the implications, limitations, and directions for future research, this is where they would look.

This section has been restructured indicating objectives, main findings, implications, limitations and future lines of research

Reviewer 2 Report

 clarified especially in the introductory chapter.

It is important to talk about sports in the context of emotional intelligence throughout the introductory / literature review chapter and not just at the end of the chapter.

It does not seem right to me to mention the way of measuring emotional intelligence in the literature review chapter, it is possible to present the theoretical concept behind the specific tool, but not the tool itself that should be presented in the method chapter (see line 33) It is important to give place to the study population (students) as well as to the cultural context (Spain), it is very possible that measuring the variables in another culture, or in another age group would have yielded different findings.

The findings chapter is beautifully written.

In the discussion section, the limitations of the study should be addressed.

Author Response

  1. It is important to talk about sports in the context of emotional intelligence throughout the introduction/literature review chapter and not just at the end of the chapter.

A total of five new references have been added in the introduction as an allusion to previous research related to this topic

  1. I do not think it is correct to mention the way to measure emotional intelligence in the literature review chapter, it is possible to present the theoretical concept behind the specific tool, but not the tool itself that should be presented in the method chapter (see line 33)

This information has been added in the instruments section (lines 238-242)

  1. It is important to give rise to the study population (students) as well as the cultural context (Spain), it is very possible that the measurement of the variables in another culture, or in another age group would have yielded different results.

Information has been added related to the study population, the fact that they are students, and that they belong to a Spanish public university (lines 228-229).

  1. In the discussion section, the limitations of the study should be addressed.

This section has been restructured indicating objectives, main findings, implications, limitations and future lines of research

Reviewer 3 Report

The paper is very concise and focused on a specific issue i.e. the relation between emotional intelligence and motivational mediator. The results are limited and might be contested but there are interesting enough to be presented as preliminary results. It has a clear statistical methodology and gets to some interesting conclusions. However, some small issues need to be addressed:

-The introduction needs a little more insight into the theoretical background of the two topics that are correlated. And especially in relation to athletes/or students, in general: is emotional intelligence important in sports/students' wellbeing? What is the role of motivation in performance? There is a lot of literature on these topics. 

- given the fact that the sample might not be significant enough  (165 students intraid of 200) to draw some general conclusions I think the title or at least the abstract should acknowledge that the results are representative only of the case study and not necessary in general.

- more contextualization of the results and discussion in the context of existing literature would give more value to this paper which is rather limited in scope and phase some "vulnerable" results. 

Author Response

  1. The introduction needs a little more information about the theoretical background of the two topics that are correlated. And especially in relation to athletes/or students, in general: is emotional intelligence important in the sports well-being of students?

The introduction has been revised, restructured and reworked. Incorporating a total of three research questions and four hypotheses

  1. What is the role of motivation in performance?

Added this information in the introduction

  1. given the fact that the sample might not be significant enough (165 intraid students out of 200) to draw some general conclusions, I think the title or at least the abstract should recognize that the results are representative only of the case study and are not necessary overall.

Added this clarification in the summary

  1. a greater contextualization of the results and discussion in the context of the existing literature would give more value to this document, which has a rather limited scope and phase for some "vulnerable" outcomes.

This section has been restructured indicating objectives, main findings, implications, comments and future lines of research

Round 2

Reviewer 1 Report

I appreciate the efforts the authors have undertaken to revise this manuscript. However, the manuscript is so replete with linguistic and stylistic errors that the substantive arguments are undermined. This paper needs careful and in-depth copy editing before I can adequately review the substantive revisions. The authors must take the stylistic elements as seriously as the substantive components. 

Author Response

Dear Editor,
Thank you for your comments and please find attached the revised and template document.
